# Systematic Genome-Wide Study and Expression Analysis of *SWEET* Gene Family: Sugar Transporter Family Contributes to Biotic and Abiotic Stimuli in Watermelon

**DOI:** 10.3390/ijms22168407

**Published:** 2021-08-05

**Authors:** Changqing Xuan, Guangpu Lan, Fengfei Si, Zhilong Zeng, Chunxia Wang, Vivek Yadav, Chunhua Wei, Xian Zhang

**Affiliations:** 1State Key Laboratory of Crop Stress Biology in Arid Areas, College of Horticulture, Northwest A & F University, Yangling 712100, China; xuanchangqing@nwafu.edu.cn (C.X.); guangpulan@nwafu.edu.cn (G.L.); sifengfei@nwafu.edu.cn (F.S.); zhengzhilong@nwafu.edu.cn (Z.Z.); chunxiawang@nwafu.edu.cn (C.W.); vivekyadav@nwafu.edu.cn (V.Y.); 2State Key Laboratory of Vegetable Germplasm Innovation, Tianjin 300384, China

**Keywords:** *Citrullus* *lanatus* (Thunb.), SWEET, phylogenetic analysis, expression analyses, plant–pathogen interaction, abiotic stresses

## Abstract

The SWEET (Sugars Will Eventually be Exported Transporter) proteins are a novel family of sugar transporters that play key roles in sugar efflux, signal transduction, plant growth and development, plant–pathogen interactions, and stress tolerance. In this study, 22 *ClaSWEET* genes were identified in *Citrullus lanatus* (Thunb.) through homology searches and classified into four groups by phylogenetic analysis. The genes with similar structures, conserved domains, and motifs were clustered into the same groups. Further analysis of the gene promoter regions uncovered various growth, development, and biotic and abiotic stress responsive *cis*-regulatory elements. Tissue-specific analysis showed most of the genes were highly expressed in male flowers and the roots of cultivated varieties and wild cultivars. In addition, qRT-PCR results further imply that *ClaSWEET* proteins might be involved in resistance to *Fusarium oxysporum* infection. Moreover, a significantly higher expression level of these genes under various abiotic stresses suggests its multifaceted role in mediating plant responses to drought, salt, and low-temperature stress. The genome-wide characterization and phylogenetic analysis of *ClaSWEET* genes, together with the expression patterns in different tissues and stimuli, lays a solid foundation for future research into their molecular function in watermelon developmental processes and responses to biotic and abiotic stresses.

## 1. Introduction

As crucially important products of photosynthesis, sugars are the predominant energy source for living chlorophytes and the source of the carbon skeletons supporting their vegetative and reproductive growth [1]. After being synthesized in photosynthetic organs, sugars are transported to storage cells by loading and unloading systems of phloem to provide the substrate of sugar metabolism, which is at the center of all biological metabolic pathway-related proteins, lipids, nucleic acids, and secondary substances [2,3]. Sugars are not only involved in the metabolic process but also act as key elements of osmotic regulation, signal identification, transient energy storage, molecule transport, and stress resistance in plants [4,5]. However, sugars cannot cross the plant bio-membrane system without the assistance of corresponding sugar transporters. These sugar transporters act as bridges that mediate the distribution of sugar between source–sink organs, the exchange of energy, and carbon in multicellular organisms [2,6]. 

So far, three eukaryotic sugar transporter families, including SWEET (Sugars Will Eventually be Exported Transporter), sucrose transporters (SUTs), and monosaccharide transporters (MSTs), have been identified in plants [7,8,9,10,11]. Unlike MSTs and SUTs, which contain twelve transmembrane domains (TMDs) and belong to the major facilitator superfamily (MFS), SWEET proteins harbor seven TMDs and catalyze fructose, hexose, and sucrose efflux along a concentration gradient that is independent of the proton gradient and pH [12,13,14]. *SWEET* genes have been identified in various plant species, including *Arabidopsis*
*thaliana* [10], rice (*Oryza sativa*) [14], wheat (*Triticum aestivum*) [15], cucumber (*Cucumis sativus*) [16], cabbage (*Brassica oleracea*) [17], rubber tree (*Hevea brasiliensis*) [18], tomato (*Solanum lycopersicum*) [19], sorghum (*Sorghum bicolor*) [20], grape (*Vitis vinifera*) [21], apple (*Malus domestica*) [22], and soybean (*Glycine max*) [23]. All *SWEET* gene members are phylogenetically divided into four groups based on the functional characterization of *SWEET* genes in *Arabidopsis*. Group III is mainly involved in sucrose uptake, while groups I, II, and IV predominantly transport monosaccharides. However, members of the same clade can localize in various cellular compartments, including the tonoplast, Golgi membrane, plasma membrane, and chloroplast [24].

The function of SWEET proteins is not limited to the diffusion of sugars across the plasma membrane or intracellular region. They are also involved in multiple physiological processes related to environmental adaptation, senescence, reproductive development, and host–pathogen interactions in higher plants [14,25]. In rice, RNA interference (RNAi) of *OsSWEET11* leads to a lower fertility rate in plants, indicating a role of this gene in pollen development. *OsSWEET14* is also found to be associated with seed development in rice. *OsSWEET14*-knockout plants showed delayed growth and smaller seed size [25]. In contrast, *OsSWEET5* overexpression lines showed growth reduction at the seedling stage compared to wild-type plants, which also displayed a premature senescence phenotype [26]. In *Arabidopsis*, *AtSWEET4* is mainly expressed in the stele of roots, leaf veins, and flowers. The protein acts as a hexose facilitator and affects plant development by influencing sugar transport to axial sinks. *AtSWEET4*-overexpression lines had higher fructose and glucose accumulation levels, along with increased plant height, while mutant lines showed opposite phenotypes with lower fructose and glucose contents and restricted plant height. Moreover, lower chlorophyll content and chlorosis in leaves was observed [27]. The *AtSWEET11*, *AtSWEET12*, and *AtSWEET15* transcripts have higher expression levels in the seed coat, and triple-knockout plants were found to have a developmental disorder of the embryo and a wrinkled seed phenotype on account of their lower starch and lipid contents, suggesting these genes play a pivotal role in seed development. Proteins encoded by *GmSWEET10a*, *10b*, *15a*, *15b*, and *ZmSWEET4c* are also involved in the development of seeds in soybean and maize, respectively [28,29,30,31]. In *Arabidopsis*, many SWEET genes, including *AtSWEET1*, *AtSWEET4–5*, *AtSWEET7-8*, and *AtSWEET13–15*, exhibited relatively higher transcript accumulation in flowers than in other tissues or organs, suggesting they are mainly involved in inflorescence development [32,33,34]. *AtSWEET9* facilitates sugar efflux in the nectary parenchyma, whereas mutant plants exhibited a loss of nectar secretion [35]. *AtSWEET10* was shown to be involved in floral transition in a photoperiod-dependent manner, while overexpression lines showed an early flowering phenotype [36]. 

*OsSWEET11* and *OsSWEET14* are negative regulators during the interaction between rice and bacterial blight (*Xanthomonas oryzae* pv. *oryzae* (*Xoo*)), and their transcript levels increased dramatically after *Xoo* infection [10,37]. In contrast, *IbSWEET10*-overexpressing plants showed more resistance to *F. oxysporum* in sweet potato, but RNAi plants showed a sensitive phenotype [38]. *AtSWEET2* plays a role in preventing the loss of sugar from root tissue. Overexpression of *AtSWEET2* reduces tolerance to *Pythium* spp. infection [39]. On the other hand, *AtSWEET4* and *AtSWEET15* have been shown to have pivotal roles in the development of clubroot disease caused by the biotrophic protist *Plasmodiophora brassicae* [27,40]. *AtSWEET2*, *4*, *7-8*, *10*, *12*, and *15* were found to be induced in *Arabidopsis* during infection with *Pseudomonas syringae* pv. *tomato* strain DC3000, and among them, *AtSWEET4*, *AtSWEET15*, and *AtSWEET17* were upregulated after *Botrytis cinerea* infection [10]. Similarly, *AtSWEET11* and *AtSWEET12* genes exhibited higher expression after *Golovinomyces cichoracearum* infection [10,41]. These findings revealed that *SWEET* proteins can be induced by pathogens, and they seem to be involved in plant–pathogen interactions.

Sugar transporters are also involved in plant responses to phytohormones and abiotic stresses. The *OsSWEET3a* protein has a dual function in rice as a gibberellin (GA) and glucose transporter [42]. Both knockout and overexpression lines of *OsSWEET3a* showed defects in germination and early shoot development, and this phenotype can be restored by exogenous gibberellin application. This suggests that *OsSWEET3a* affects plant growth and development via a gibberellin-mediated response. *BnSWEET12* is induced by brassinosteroid (BR), GA, and abscisic acid biosynthetic (ABA) in oilseed rape [43]. In *Arabidopsis*, *AtSWEET13* and 14 are transporters of GA, and their double-mutant lines cannot transport exogenous GA and showed altered responses to GA during germination at the seedling stage [44]. Salinity and drought stress activate ABA genes and increase ABA content, which in turn activates ABA-responsive genes through ABA-responsive elements. The promoters of *OsSWEET13* and *OsSWEET15* harbor a site for the ABA-responsive transcription factor OsbZIP72, which can activate their expression under drought and salinity stresses. This mechanism potentially modulates sucrose distribution and transport in response to abiotic stress [45]. *AtSWEET4* plays an important role in plant freezing tolerance. *AtSWEET4*-silenced lines showed susceptibility to freezing, and *AtSWEET4*-overexpression lines have higher freezing tolerance [27]. In previous research, *AtSWEET11* and *12* were also found to participate in cold tolerance; compared to wild-type plants, *AtSWEET11* and *12* mutants exhibited greater freezing tolerance. Moreover, *AtSWEET11* and *AtSWEET12* also exhibited higher expression in plants under water deficit, and it was assumed that this phenomenon could be related to an increase in carbon export from the leaves to the roots [46,47]. *AtSWEET15* can be induced by osmotic, drought, salinity, and cold stresses via the ABA-dependent pathway, and *AtSWEET15*-overexpressing plants showed accelerated senescence and reduced cell viability in the root. In addition, the corresponding overexpression plants were hypersensitive to cold and salinity stress [48]. Cold stress also induced the transcription of *CsSWEET1a* and *CsSWEET17* in the tea plant [49]. Furthermore, the *CaSWEET16* and *AtSWEET16* genes are repressed during cold acclimation, and plants that overexpressed these genes showed higher freezing tolerance [50,51].

Watermelon (*Citrullus lanatus* (Thunb.)) is an economically important horticulture crop widely grown for its edible fruit. Sugars play an important role in watermelon fruit quality and economic value, as they are the main photosynthetic products stored in the fruit by sugar transporters. The *SWEET* genes have been demonstrated to play important roles in plant growth and plant–pathogen and plant–environment interactions in many species, but they have not been studied in watermelon. In this study, we have conducted the first genome-wide analysis of *SWEET* genes in watermelon, named *ClaSWEET* genes, and analyzed their chromosomal distribution, gene structure, motif distribution, phylogenetic relationships, *cis*-regulatory elements, spatial and temporal expression patterns, and expression in response to biotic and abiotic stress. The present study will provide insights for future research on *ClaSWEET* genes associated with growth, development, and various stress responses in watermelon.

## 2. Results

### 2.1. Identification, Annotation, and Uneven Chromosomal Distribution of Watermelon SWEET Genes

A total of 22 *ClaSWEET* genes were identified in watermelon using the homologous sequences of *Arabidopsis* as queries. The *ClaSWEET* genes were identified and named from *ClaSWEET1* to *ClaSWEET22* based on their chromosome locations, and each *AtSWEET* gene corresponds to approximately one to five *ClaSWEET* genes (Table 1). These *ClaSWEET* genes are distributed over the majority of watermelon chromosomes, except for chromosomes 2, 4, and 9. The highest number of genes were mapped to chromosome 1, whereas chromosomes 5 and 7 each contained just one *ClaSWEET* gene. Chromosomes 3, 6, and 11 contained three *ClaSWEET* genes, and chromosomes 8 and 10 contained two and four *ClaSWEET* genes, respectively (Figure 1). The coding domain sequences of the *ClaSWEET* genes ranged from 207 to 900 bp in length, and inferred peptide lengths varied from 68 to 299 aa. All *ClaSWEET* members except *ClaSWEET4*, *5*, *7*, and *21* had theoretical isoelectric point values above 7.0, whereas overall, theoretical isoelectric point values ranged from 4.46 to 9.83. The molecular weights ranged from 7.66 to 33.68 kDa. Subcellular localization prediction indicated that proteins encoded by *ClaSWEET* genes function on the plasma membrane, extracellular, cytoplasmic, endoplasmic reticulum, tonoplast membrane, and chloroplast thylakoid membrane, respectively (Table 1).

### 2.2. Phylogenetic and Conserved Domain Analysis of ClaSWEET Genes

A phylogenetic tree consisting of *Arabidopsis*, cucumber, rice, and watermelon SWEETs was constructed by the neighbor-joining (NJ) method to elucidate the evolutionary relationship of this family (Figure 2). Based on the classification of *Arabidopsis*, all of the SWEET proteins were categorized into four discrete clades. The largest clade (II) consisted of five *AtSWEET* proteins, seven *OsSWEET* proteins, five *CsSWEET* proteins, and ten *ClaSWEET* proteins. Seven *AtSWEET* proteins, five *OsSWEET* proteins, six *CsSWEET* proteins, and seven *ClaSWEET* proteins were confined to the second-largest clade (clade III). Clade I contained six *OsSWEET* proteins and nine proteins for *Arabidopsis*, cucumber, and watermelon (three for each species, respectively), whereas clade IV was the smallest clade and included two *AtSWEET*, one *OsSWEET*, two *ClaSWEET*, and three *CsSWEET* proteins. 

The putative protein sequences of *ClaSWEET* genes were aligned to analyze the conserved MtN3/saliva regions of watermelon SWEET members. As shown in Figure 3, most *ClaSWEET* proteins were predicted to harbor two MtN3/saliva domains. However, five of twenty-two SWEET proteins (*ClaSWEET4*, *5*, *18*, *20*, and *21*) had only one MtN3/saliva domain. Additionally, four serine and two tyrosine sites, as well as one threonine phosphorylation site, were predicted in the two conserved MtN3/saliva regions. A total of seven motifs were identified among the *ClaSWEET* members after their sequences were analyzed using the MEME motif program (Figure 4b). Motif 6, as a part of the second conserved MtN3/saliva domain, was observed in all putative *ClaSWEET* proteins, while motif 7 was mainly predicted in the members of clade II. Notably, in clade II, the majority of members (*ClaSWEET2*, *3*, *13*, *15*, *16*, and *19*) contained all seven motifs, in contrast to *ClaSWEET**4* and *5*, which each had only one motif 6.

### 2.3. Gene Structure and Cis-Regulatory Elements Prediction of ClaSWEET Genes

Based on the phylogenetic relationships of *ClaSWEET* genes (Figure 4a), their gene structure and intron phases were characterized (Figure 4c). Here, 12 out of 22 members had six exons (*ClaSWEET3*, *6–10*, *14–17*, *19*, and *22*), and they had the same intron phase patterns except for *ClaSWEET14* and *17*. *ClaSWEET2*, *11*, and *13* contained five exons. In contrast, *ClaSWEET1*, *12*, and *18* contained four exons, and *ClaSWEET4* had three exons. *ClaSWEET5*, *20* and *21* harbored just two exons. In addition, gene pairs in the sister branch were generally identified to have similar structural features, such as *ClaSWEET2* and *ClaSWEET13*, *ClaSWEET8* and *ClaSWEET9*, and *ClaSWEET20* and *ClaSWEET21*, as each pair had similar exon and intron numbers and CDS lengths. 

To investigate the responses to various factors by *ClaSWEET* members, the 2.5 kb sequences upstream of the ATG start codon of these genes were submitted to the PlantCARE server to predict their promoter *cis*-regulatory elements. The 26 *cis*-regulatory promoter elements related to phytohormones, stress, growth, and development that were identified are shown in Figure 5. The phytosterols-responsive *cis*-regulatory elements included one abscisic-acid-responsive element (ABRE), two auxin-responsive elements (AuxRR-core and TGA-element), two methyl-jasmonate-responsive elements (TGACG and CGTCA motifs), one ethylene-responsive element (ERE), three gibberellin-responsive elements (TATC-box, P-box, and GARE-motif), and one salicylic-acid-responsive element (TCA-element). The stress-responsive *cis*-regulatory elements consisted of one anaerobic-responsive element (ARE), one low-temperature-responsive element (LRT), two drought-responsive elements (MBS and MYC), one wound-responsive element (WUN-motif), and three defense- and stress-responsive elements (MYB, W-box, and TC-rich repeats). Additionally, there were five *cis*-regulatory elements related to meristem expression (CAT-box) and involved in circadian control (circadian), differentiation of palisade mesophyll cells (HD-Zip I), cell cycle regulation (MSA-like), and seed-specific regulation (RY-element). Moreover, three light-responsive elements (G-box, GT1-motif, and AAAC-motif) were found in the promoter regions. Most of the *ClaSWEET* genes had one or more *cis*-regulatory elements related to hormone- and stress-related functions, suggesting that *ClaSWEET* proteins are involved in multiple physiological processes through various environmental adaptations.

### 2.4. Expression Patterns of ClaSWEET Genes in Cultivars and Wild Varieties of Watermelon

To investigate the transcriptional levels of *ClaSWEET* genes in different organs, the roots, stems, leaves, tendrils, and female and male flowers of one cultivated and two wild watermelon varieties were collected to assess the expression patterns of these gene members by qRT-PCR. The heatmap shown in Figure 6 reflects the global expression patterns and the clustering of *ClaSWEET* genes. Almost all the members (except *ClaSWEET4*, *5*, *10*, and *15–17* in ‘Sugar baby’; *ClaSWEET4*, *15*, and *17* in ‘PI296341-FR’; *ClaSWEET4* and *15–17* in ‘YL’) were detected in more than one of the tested tissues. In ‘Sugar baby,’ five *ClaSWEET* genes (*ClaSWEET1*, *6*, *9*, *13*, and *18*) in roots, seven *ClaSWEET* genes (*ClaSWEET2*, *7*, *8*, *11*, *14*, *20*, and *21*) in female flowers, and all *ClaSWEET* genes (except *ClaSWEET4–6*, *10*, and *15–18*) in male flowers had a significant accumulation of mRNA (Figure 6a). Notably, *ClaSWEET6* and *18* were particularly highly expressed in the roots, whereas *ClaSWEET19* and *22* exhibited elevated expression only in the male flowers of ‘Sugar baby’ plants.

In ‘PI296341-FR,’ fifteen members (*ClaSWEET3*, *6–**14*, *16*, *18-20*, and *22*) in roots, three members (*ClaSWEET3*, *9*, and *20*) in both stems and leaves, two members in tendrils (*ClaSWEET3* and *20*), two members in female flowers (*ClaSWEET11* and *14*), and sixteen members (*ClaSWEET1–3*, *5*, *7–14*, and *19-22*) in male flowers were highly expressed (Figure 6b). *ClaSWEET16* and *18* showed higher transcript abundances just in the roots, and *ClaSWEET1*, *2*, *5*, *10*, and *21* were specifically expressed in female flowers and not other organs. 

Moreover, most of the *ClaSWEET* genes (*ClaSWEET1–3*, *5–14*, and *19–22*) showed high transcript abundances in male flowers of ‘YL’ plants (Figure 6c). *ClaSWEET18* showed preferential expression in the roots, and *ClaSWEET7* was preferentially expressed in the stems. In contrast, *ClaSWEET2*, *5*, *10–12*, *19*, *21*, and *22* displayed high transcriptional levels only in male flower tissue.

### 2.5. Expression Patterns of SWEET Genes in Response to Stress Treatment with Fon Infection

To identify the involvement of *ClaSWEET* gene responses to biotic stress, cultivar and wild varieties of watermelon that are susceptible and resistant to Fusarium wilt (‘Sugar baby’ and ‘PI296341-FR’, respectively) were infected with Fusarium wilt (*Fusarium oxysporum* f. sp. *niveum* (Fon)), and their roots and stems were independently sampled after being infected. In the root tissue of the susceptible ‘Sugar baby’ cultivar, 19 out of 22 *ClaSWEET* genes were transcriptionally upregulated compared to the control; in contrast, the mRNA levels of *ClaSWEET1*, *ClaSWEET2*, and *ClaSWEET15* were lower (Figure 7a). A portion of *ClaSWEET* members (*ClaSWEET3–8*, *10*, *13*, and *14*) were induced in the roots of the resistant ‘PI296341-FR’ cultivar; in contrast, 13 members, including *ClaSWEET1*, *2*, *9*, *11*, *12*, and *15–22*, showed downregulated expression. Furthermore, the Fusarium wilt infection highly induced transcription of 14 *ClaSWEET* genes (*ClaSWEET4*, *7*, *9–15*, *17*, and *19–22*) in the stems of the susceptible ‘Sugar baby’ cultivar and of 10 *ClaSWEET* genes (*ClaSWEET3*, *5*, *6*, *9–12*, *14*, *17*, and *22*) in stems of the resistant ‘PI296341-FR’ cultivar. In the stem, there was a slight increase in *ClaSWEET7* and *20* transcripts in the resistant ‘PI296341-FR’ cultivar after infection, but more mRNA accumulation compared to the control in the susceptible ‘Sugar baby’ cultivar (Figure 7b). Notably, the relative expression of *ClaSWEET9*, *11*, *12*, and *16–22* increased multifold over the control in the root tissues of the susceptible ‘Sugar baby’ cultivar; however, the relative expression of these genes decreased compared with the control in the resistant ‘PI296341-FR’ cultivar (Figure 7c). Thus, resistant and susceptible cultivars showed differential expression of *ClaSWEET* genes, indicating that these genes could play an important role in host–pathogen interactions.

### 2.6. Induced Expression of SWEET Genes in Response to Low-Temperature, Salt, and Drought Stress

Watermelon seedlings treated with abiotic stress conditions (drought, high salinity, and low temperature) were used to study the differential expression pattern of *ClaSWEET* genes. Under drought conditions, the majority of genes observed exhibited slightly increased expression regardless of drought duration, and only four genes, including *ClaSWEET1*, *15*, *21*, and *22*, were downregulated after two days of stress conditions (Figure 8a). Moreover, with the increase in drought treatment time, all the *ClaSWEET* genes showed a tendency to accumulate mRNA content, except for *ClaSWEET1*, *13*, *14*, *21*, and *22*, which were found to have low mRNA levels at 6 dpt compared to the control. At 8 dpt, all the members exhibited significantly up-regulated expression. To obtain insight into the underlying functional roles of *ClaSWEET* genes in the response to salt stress, the expression patterns of the *ClaSWEET* members were analyzed after salinity stress (Figure 8b). During the salt stress treatment, the expression of *ClaSWEET1* showed lower expression, while ten *ClaSWEET* genes (*ClaSWEET2*, *6–8*, *10*, *13*, and *15–18*) were significantly upregulated. The transcription of *ClaSWEET14* and *22* showed an obvious decrease at 6 hpt but increased expression was observed at a later time post treatment. The transcriptional levels of *ClaSWEET* members were studied to elucidate the mechanism of gene responses to low-temperature stress (Figure 8c). Thus, ten members were identified to respond to low temperature with increased expression levels, *ClaSWEET2*, *4*, *11*, *13–17*, *21*, and *22*. Ten *ClaSWEET* genes were downregulated at 6 hpt (*ClaSWEET3*, *5–7*, *9*, *10*, *12*, and *18–20*), and their expression increased afterwards. The expression of *ClaSWEET8* showed an obvious increase at 6 and 12 hpt, but a decrease at 24 and 48 hpt, which contrasted with the expression of *ClaSWEET20*. These expression results under various abiotic stresses suggest that *ClaSWEET* genes act as an important regulator of plant responses.

## 3. Discussion

The *SWEET* gene family has been characterized as encoding sugar transporters that mainly function in sugar efflux [10]. Despite their sugar efflux function, a portion of SWEET family proteins have been identified to be involved in reproductive development, senescence, host–microbe interactions, and abiotic stress responses in many plant species [14,26,38,39,47,48]. Although watermelon is one of the most economically important and popular fruit crops, systematic studies on *SWEET* homologs in watermelon have not been reported.

In this study, 22 *SWEET* genes were identified in watermelon through a genome-wide search, and these genes were distributed over eight chromosomes (Figure 1). According to the evolutionary relationship inferred by phylogenetic analysis, these genes were classified into four clades (Figure 2), consistent among *Arabidopsis*, rice, and cucumber [10,16]. It was reported that *AtSWEET* members that fall in clade I (*AtSWEET*1–3), II (*AtSWEET*4–8), and IV (*AtSWEET16* and *17*) preferentially transport monosaccharides, and members of clade III (*AtSWEET*9-15) are involved in sucrose transport [4]. In addition, the SWEET proteins in clade IV are able to control the flux of fructose across the tonoplast [10,50,52]. In *Arabidopsis*, *AtSWEET*16 and *AtSWEET*17 are located on the tonoplast membrane and act as fructose transporters. Similarly, the homologs *ClaSWEET8* and *9* in watermelon also fall into clade IV, and subcellular prediction revealed the presence of these genes in the plasma membrane and tonoplast membrane, respectively, suggesting *ClaSWEET8* and *ClaSWEET9* proteins might have similar functions as *AtSWEET*16 and *AtSWEET*17 in *Arabidopsis*. 

In different species of monocots and dicots, SWEET proteins were predicted to consist of seven TMDs that harbor two MtN3/saliva domains [10]. However, the SWEET proteins in prokaryotes only contain three TMDs that harbor just one MtN3/saliva domain, suggesting that during the evolutionary process, a replication fusion occurred in one MtN3/saliva domain of prokaryotes, resulting in the generation of a SWEET protein with two MtN3/saliva domains in eukaryotes [14,53]. The sequence comparison analysis revealed that 15 *ClaSWEET* proteins (about 68%) had two completely conserved MtN3/saliva domains, and the rest of the members harbored one or one and a half MtN3/saliva domains (Figure 3). Similar arrangements of domains have also been observed in rice and sorghum, where *OsSWEET7a*, *OsSWEET7e*, Os01g40960, and SbSWEET3–4 each had just one MtN3/saliva domain [14]. These short SWEET proteins may have been formed by tandem and domain duplication events that occurred during evolution. Four serine and two tyrosine sites, as well as one threonine phosphorylation site, were observed in the conserved domains, out of which the two serine sites and one threonine phosphorylation site were located on the inner sides of the membrane and one tyrosine phosphorylation site was located on the outer sides (Figure 3). These findings indicate that domains of *ClaSWEET* proteins that are located on both sides of the membrane could be important functional regions. The phosphorylation sites located on both sides of the membrane are probably related to signal recognition and transduction functions of *ClaSWEETs*.

In *Medicago truncatula* [25], cucumber [16], banana [13], *Brassica rapa* [6], cabbage [17], tomato [19], soybean [23], rubber tree [18], cotton [54], and pear [55], more than half of the *SWEET* members have six exons and five introns, suggesting that during evolution the molecular features of *SWEET* genes were highly conserved. Here, the gene structure analysis of *ClaSWEET* genes indicated that 12 members (about 54.5%) harbored six exons and five introns. Moreover, the *ClaSWEET* members in the sister branch may have similar functions owing to their high structural similarity. It has also been reported that after segmental duplication, intron loss is faster than intron gain [56]. We propose that *ClaSWEET* genes with the maximum number of introns are the original homologs of those members with fewer introns, i.e., those exhibiting potential intron loss. 

Higher transcriptional levels of *ClaSWEET* genes were observed in flowers (especially in male flowers) than in other organs or tissues (Figure 6) in both wild varieties and cultivated cultivars. Consistent findings were reported in other species [10,15,16]. In contrast, the spatial expression in various watermelon cultivars showed that *ClaSWEET18* was highly expressed in root tissue, whereas its homolog *AtSWEET**5* in *Arabidopsis* was highly expressed in gametophytes [57]. Additionally, *AtSWEET7* was preferentially expressed during pollen development [57], while the homolog *ClaSWEET13* was mainly expressed in roots and male flowers, suggesting that *ClaSWEET13* is not only associated with root growth but also reproductive organ development. 

In order to grow and reproduce, many pathogens have evolved mechanisms to acquire glucose from their hosts by hijacking their sugar efflux systems; thus, pathogens are able to alter sugar efflux at the site of infection and modulate plant immunity [10]. For example, six *BoSWEET* genes were expressed at higher levels in the roots of susceptible cabbage plants, but none were expressed at higher levels in resistant lines after *P. brassicae* infection, indicating these genes could be associated with the response to *P. brassicae* colonization [17]. Fusarium wilt is a major soil-borne disease caused by Fon, which colonizes the extravascular system of root and stem tissues and seriously devastates watermelon crop production worldwide [58,59]. It is unsurprising that the *ClaSWEET* genes displayed differential expression patterns between susceptible and resistant cultivars after infection (Figure 7a). In the roots of the susceptible cultivar, nineteen *ClaSWEET* genes were upregulated and three genes were downregulated. In contrast, in the resistant cultivar, nine *ClaSWEET*s were upregulated, and thirteen genes were downregulated. *ClaSWEET9*, *11*, *12*, and *16–22*, however, showed variety-specific downregulation in resistant cultivar roots, though their transcriptional expression was very high in the control (Figure 7c). Moreover, the relative expression of *ClaSWEET15*, *19*, and *21* was also downregulated in the stem tissue of the resistant cultivar, while their expression was induced in the stem tissue of the susceptible cultivar (Figure 7b). Thus, we hypothesize that *ClaSWEET7*, *9*, *11*, *12*, and *15–22* may play key roles in reducing the sugar efflux to inhibit the growth of *Fusarium oxysporum*. 

Plants have evolved sensory and response mechanisms that allow them to physiologically adapt to environmental stresses under adverse conditions such as drought, high salinity, and low-temperature stress. Sugars, as osmo-protectants and molecular switches, can regulate resistance and the adaptability of plants under stress [60]. Previous research has revealed that *AtSWEET4*, *AtSWEET11*, *AtSWEET12*, *AtSWEET15*, *AtSWEET16*, and *AtSWEET17* can respond to a variety of abiotic stresses in *Arabidopsis* [27,46,47,50,52]. The transcriptional levels of five *BoSWEET* members were downregulated under chilling stress in cabbage [17]. Similarly, under various stresses, the mRNA levels of six *GhSWEET* genes showed significant upregulation in cotton. In the present study, most *ClaSWEET* genes showed relatively higher expression under stress conditions, suggesting *ClaSWEET* proteins are involved in the response of plants to stresses. Eight *ClaSWEET* genes (*ClaSWEET3*, *5*, *9*, *12*, and *18–20*) showed an initial decrease in expression followed by increased expression at 24 and 48 hpt under salt and low-temperature stress, respectively. This may be owing to the different sensitivities of these genes to salt damage and low-temperature stress. Similar findings were observed when Kentucky bluegrass was artificially exposed to low-temperature stress; the expression of *PpSWEET1a* and *PpSWEET17* showed decreased expression during the initial stage of stress, followed by elevated expression at a later stage [2]. On the other hand, the transcriptional levels of two *ClaSWEET* genes (*ClaSWEET16* and *17*) were markedly increased under three treatments throughout the whole stage, suggesting that *ClaSWEET16* and *ClaSWEET17* may also have the potential to respond to other stresses. 

In this article, a relatively comprehensive study of the *ClaSWEET* gene family was conducted, which may clarify the biological functions of *ClaSWEET* proteins regarding their role in developmental processes and responses to stress. However, the understanding of their exact biological function remains incomplete. In addition, it is still unknown what type of sugar is transported by *ClaSWEET* members in different clades and the roles they play in responding to phytohormones. Thus, deep functional validation research is required to provide valuable insights to aid plant engineering strategies for developing crops resistant to adverse stress conditions.

## 4. Materials and Methods

### 4.1. Sequence Retrieval and Domain Confirmation of Watermelon SWEET genes

The related sequences of published *Arabidopsis*
*thaliana*, rice, and cucumber *SWEET* genes were obtained from the *Arabidopsis* information resource (https://www.arabidopsis.org/, accessed on 2 February 2021), rice genome database (http://www.ricedata.cn/gene/, accessed on 8 February 2021), and cucumber genome database (http://cucurbitgenomics.org/organism/20, accessed on 8 February 2021). The protein sequences encoded by *AtSWEET* genes were used as query sequences to perform BLAST search in the watermelon genome database (E-value cutoff of e-10; http://cucurbitgenomics.org/organism/21, accessed on 15 March 2021). Then, putative *ClaSWEET* proteins were submitted to the Pfam database (http://pfam.xfam.org/search/sequence, accessed on 15 March 2021) and SMART database (http://smart.embl-heidelberg.de, accessed on 15 March 2021) for confirmation of conserved MtN3_saliva domain, which was further confirmed by NCBI conserved domain database (https://www.ncbi.nlm.nih.gov/cdd, accessed on 15 March 2021). According to their positions in the genome, the filtrated *ClaSWEET*s were named from *ClaSWEET1* to *ClaSWEET22* (Table 1).

### 4.2. Gene Structure, Cis-Regulatory Element, Protein Properties, and Phylogenetic Analysis

The information of chromosomal locations, intron and exon numbers, CDS, and protein sequences of *ClaSWEET* genes were acquired from the Cucurbit Genome Database. A schematic diagram of exon/intron distribution pattern and intron phases was constructed via Gene Structure Display Server (http://gsds.gao-lab.org, accessed on 10 March 2021). Furthermore, the upstream 2.5 kb sequence starting from the ATG codon was submitted to the PlantCARE Server (http://bioinformatics.psb.ugent.be/webtools/plantcare/html, accessed on 16 January 2021) to predict *cis*-regulatory elements in promoter regions. Subcellular localization prediction of each gene was predicated by WoLF PSORT (http://www.genscript.com/psort/wolf_psort.html, accessed on 12 January 2021). Physiochemical properties, including molecular weight and theoretical isoelectric point of *ClaSWEET* proteins, were predicted using the Protparam tool (http://expasy.org, accessed on 16 January 2021). TMHMM Server v.2.0 (http://www.cbs.dtu.dk/services/TMHMM, accessed on 12 January 2021) was utilized to obtain the number of transmembrane domains (TMDs). The *ClaSWEET* protein sequences alignment was performed using the Clustal Omega (http://www.ebi.ac.uk/Tools/msa/clustalo, accessed on 16 January 2021) program with the default parameters and conserved serine predicted by NetPhos 2.0 (http://www.cbs.dtu.dk/services/NetPhos, accessed on 16 January 2021). The BoxShade (http://www.ch.embnet.org/software/BOX_form.html, accessed on 10 March 2021) program was utilized to highlight conserved or similar amino acid sequences. The phylogenetic tree construction was performed by the MEGA 7.0.21 program with neighbor-joining method and 1000 bootstrap interactions test. According to the classification of *AtSWEET*s and *OsSWEETs* [10], the *ClaSWEET*s fall into four clades.

### 4.3. Plant Cultivation and Treatments

One cultivated variety (‘Sugar baby’) and two wild (‘PI296341-FR’ and ‘YL’) watermelon cultivars were seeded into 50-cell propagation trays in a controlled growth chamber under the following conditions: 26 ± 2 °C, 14 h light, and 10 h dark (day/night) photoperiod, photosynthetic photon flux density (PPFD) of 600 µmol m^−2^ s^−1^, and relative humidity of 70–90%. Fifteen-day-old seedlings of the *Fon* race 2-resistant (‘PI296341-FR’) and *Fon* race 2-susceptible (‘Sugar baby’) cultivars were prepared for inoculation [59], and 21-day-old seedlings of ‘YL’ were used to study the abiotic stresses (drought, high salinity, and low temperatures). 

*Fon* race 2 was grown on Difco™ Potato Dextrose Agar for two weeks, then the hyphae were added to Difco™ Potato Dextrose Broth and placed on a rotary shaker at 120 rpm for 10–14 days at 25 °C. The conidial concentration was adjusted to 1 × 10^6^ conidia per mL with distilled water after the spore suspension was filtered through four layers of cheesecloth [58]. The seedlings of PI296341-FR and Sugar baby were uprooted from potting soil, the roots were trimmed after being washed in water, then the seedlings were set in the conidial suspension for 10 min, whereafter, they were transplanted into a media mixture of perlite:vermiculite:potting soil (1:1:1) filled in 50-cell propagation trays. 

The 21-day-old seedlings of ‘YL’ were divided into three parts and treated with drought, high salinity, and low temperature, respectively. For drought treatment, plant seedlings were not irrigated for 8 days; their leaves were collected at 0, 2, 4, 6, and 8 days post treatment (dpt). For high-salinity treatment, seedlings were irrigated with 250 mM NaCl solution, and their leaves were sampled at 0, 6, 12, 24, and 48 h post treatment (hpt) [61]. For the low-temperature treatment, plant samples were placed in the growth chamber at 4 °C for 48 h with 14 h-light and 10 h-dark photoperiod, and the leaves were sampled at 0, 6, 12, 24, and 48 hpt [62]. All the samples collected at 0 dpt and 0 hpt were used as controls, and five roots or leaves from different plants were pooled at each time point with three biological replicates, which were immediately frozen in liquid nitrogen and stored at −80 °C until further analysis. 

### 4.4. RNA Extraction and Quantitative RT-PCR Analysis

The leaf, root, stem, tendril, female flower, and male flower samples of three watermelon varieties were collected for tissue-specific expression analysis. The root and stem samples from infected plants of ‘PI296341-FR’ and ‘Sugar baby’ were collected for biotic stress analysis. Furthermore, for the abiotic stress, the leaf samples from ‘YL’ seedlings were collected for RNA extraction. 

Total RNA of tissues or organs was extracted using RNAsimple Total RNA Kit (Tiangen, Beijing, China) following the manufacturer’s instructions and further purified by FastKing RT kit (Tiangen). The single-strand synthesis of the cDNA was carried out by FastKing RT kit (Tiangen, Beijing, China) according to the manufacturer’s instructions. SYBR^®^ Green I Master (Aidlab, Beijing, China) was used for quantitative real-time PCR with a LightCycler^®^ system (Roche Diagnostics, Shanghai, China). The specific and efficient primers of *ClaSWEET* genes were used to amplify their target genes (Appendix A). The amplification of the qRT-PCR process was executed as follows: (1) pre denaturation at 94 °C for 5 min; (2) 94 °C for 10 s, 60 °C for 30 s, and 72 °C for 30 s, for 40 cycles. Actin gene was selected as an internal control to normalize transcriptional levels. Genes and internal controls were amplified in triplicate, and relative gene expression levels were calculated using the 2^−ΔΔ^CT method [63]. The expression level of all identified *ClaSWEET* genes was log2 transformed and normalized to obtain a heatmap.

## 5. Conclusions

In the present study, twenty-two *ClaSWEET* members were retrieved from the watermelon genome, and were unevenly distributed on eight chromosomes. *ClaSWEET* members were classified into four groups according to the phylogenetic relationship of *Arabidopsis*, rice, and cucumber. Gene structures, conserved motifs, and domain patterns displayed universal similarities in sister branches or the same group, suggesting they may have an analogous function. The putative amino acid phosphorylation sites in conserved domains and *cis*-regulatory elements in promoter regions indicated that both intracellular and extracellular regions could be their functional areas and have a potential functional response to hormones and signals. In addition, the expression patterns of *ClaSWEET* genes in different tissues and relative expression level analyses in the given stresses demonstrated that *ClaSWEET* proteins play key roles in watermelon development and responses to biotic and abiotic stresses. Overall, these results lay a foundation for future studies on *ClaSWEET* gene function and explore their potential application to the improvement of biotic and abiotic stress tolerance in watermelon plants.

## Figures and Tables

**Figure 1 ijms-22-08407-f001:**
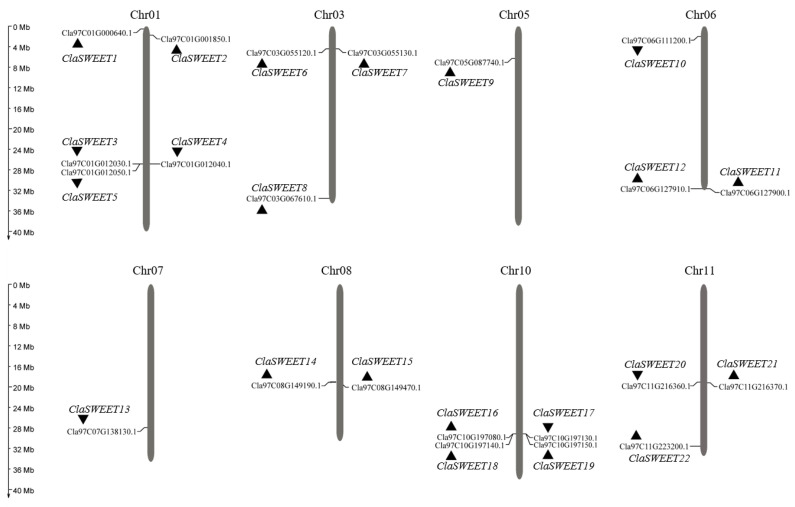
Distribution of twenty-two *ClaSWEET* genes on watermelon chromosomes. The scale ruler on the left side shows the physical distance (Mb) of the chromosomes. The relative positions of *ClaSWEET*s are marked on the chromosomes. According to their physical location on the chromosomes, the *SWEET* genes of watermelon named from *ClaSWEET1* to *ClaSWEET22* correspond to the genes’ ID. Triangles represent the transcriptional direction of each gene: ▲ backwards transcription, ▼ forward transcription.

**Figure 2 ijms-22-08407-f002:**
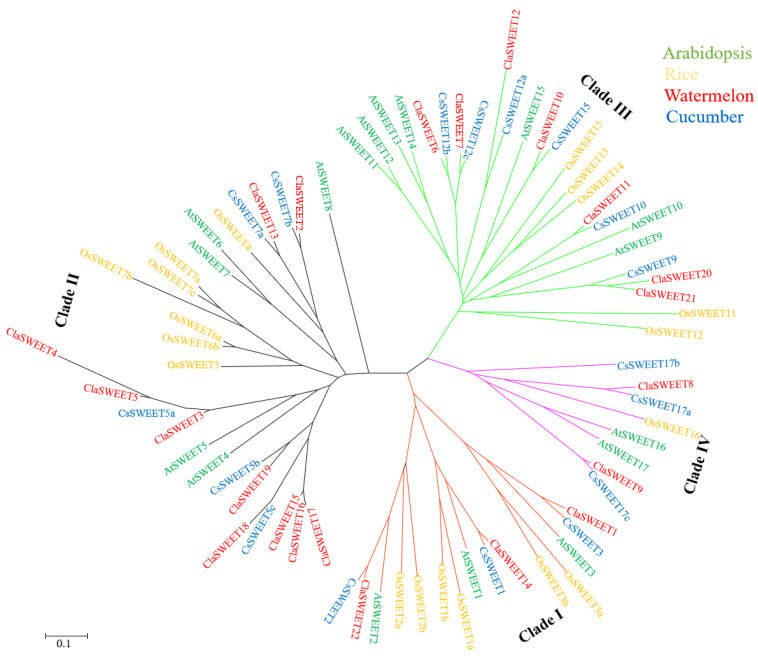
The unrooted phylogenetic tree of SWEET superfamily was generated based on the amino acid sequences of *Arabidopsis*, rice, cucumber, and watermelon by the neighbor-joining (NJ) method using MEGA 7.0.21. The SWEET members were categorized into four clades and labeled as I, II, III, and IV. The SWEET members of species were color-coded: At, *Arabidopsis* (**green**); Os, rice (**yellow**); Cs, cucumber (**blue**); Cla, watermelon (**red**).

**Figure 3 ijms-22-08407-f003:**
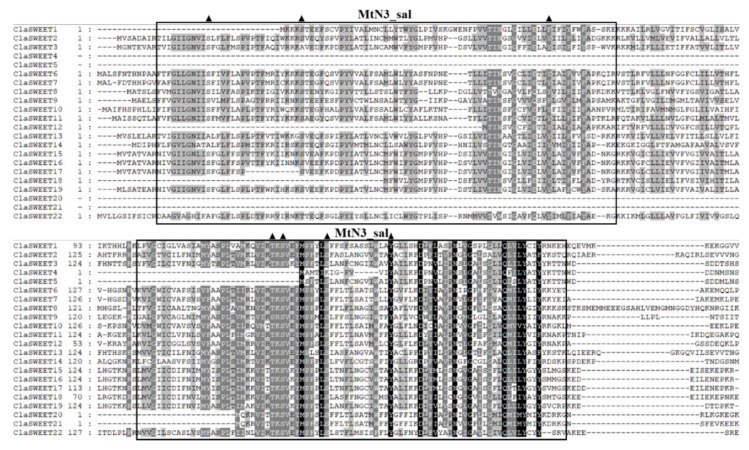
Multiple sequence alignment of the *ClaSWEETs*. The sequences contained in the black box are conserved domains unique to *ClaSWEET* members. The position of the conserved serine (S), threonine (T), and tyrosine (Y) predicted to be the phosphorylation sites are indicated by the triangles.

**Figure 4 ijms-22-08407-f004:**
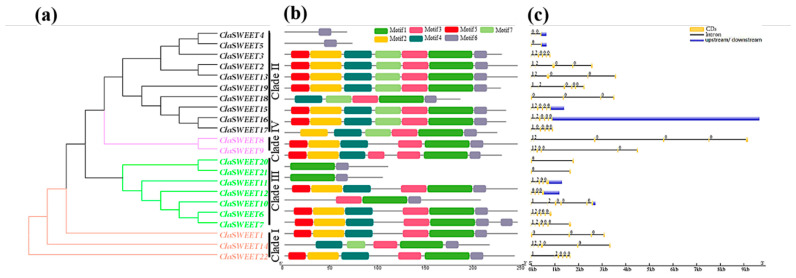
Phylogenetic relationships, distribution of conserved motifs, and structures of *ClaSWEET* genes. (**a**) The phylogenetic tree of *ClaSWEETs* using the neighbor-joining (NJ) method. (**b**) The conserved motifs of *ClaSWEET* members. The colored squares correspond to seven different conserved sequences. (**c**) Structure of *ClaSWEET* genes. Green boxes and gray lines represent exons and introns, respectively. 0: phase 0, 1: phase 1, and 2: phase 2, indicate an intron located between two complete codons, inserted at the first base of a codon and the second base of a codon, respectively.

**Figure 5 ijms-22-08407-f005:**
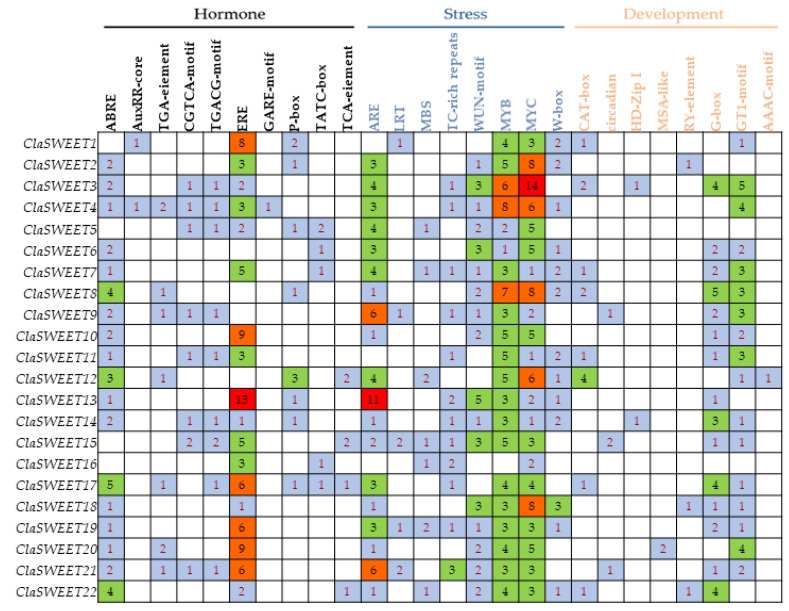
The *cis*-regulatory elements involved in plant hormone, development, and stress responses in the upstream regions of *ClaSWEET* genes. ABRE, abscisic-acid-responsive element; AuxRR-core, auxin-responsive element; TGA-element, auxin-responsive element; CGTCA-motif, TGACG-motif, MeJA-responsive elements; ERE, ethylene-responsive element; GARE-motif, P-box, and TATC-box, gibberellin-responsive elements; TCA-element and W-box, salicylic-acid-responsive elements; ARE, involved in the anaerobic induction; LTR, low-temperature-responsive element; MBS, TC-rich repeats, MYB and MYC, involved in defense and stress responsiveness; WUN-motif, wound responsiveness; CAT-box, circadian, HD-Zip I, MSA-like, and RY-element, involved in meristem expression, circadian control differentiation of the palisade mesophyll, cell cycle regulation, and seed-specific regulation, respectively. G-box, GT1-motif, and AAAC-motif are light-responsive elements.

**Figure 6 ijms-22-08407-f006:**
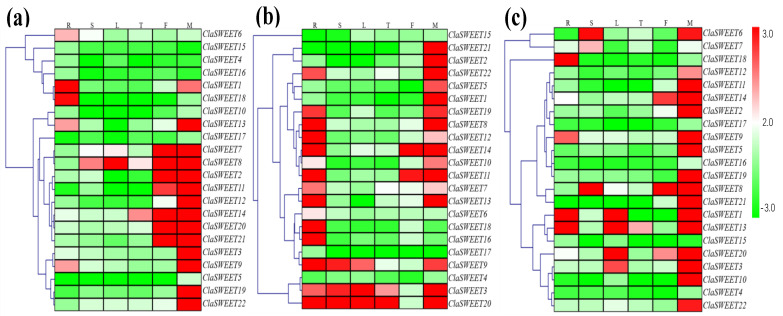
Expression profiles of tissues- or organs-specific *ClaSWEET* genes in one cultivated and two wild varieties of watermelon. (**a**) The expression profiles of cultivated variety Sugar baby. (**b**) The expression profiles of wild variety PI296341-FR. (**c**) The expression profiles of wild variety YL. The total RNA was extracted from R (roots), S (stems), L (leaves), T (tendrils), F (female flowers), and M (male flowers) of three varieties of watermelon. Red and green correspond to strong and weak expression of the *ClaSWEET* genes, respectively. The phylogenetic tree was built with average linkage clustering method.

**Figure 7 ijms-22-08407-f007:**
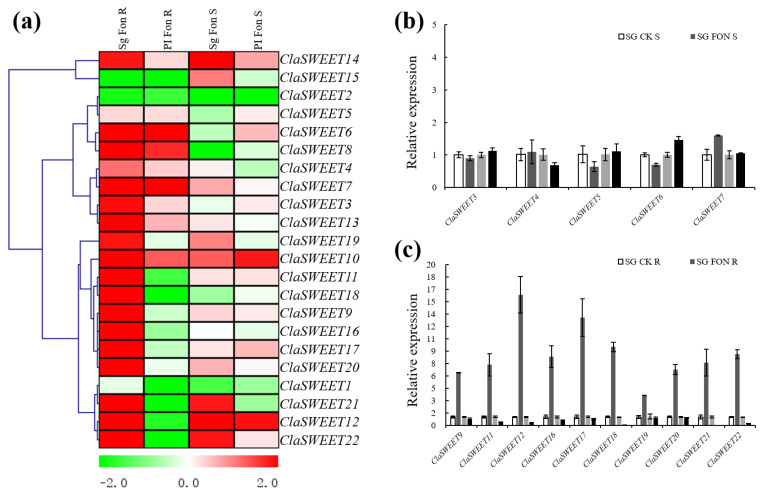
(**a**) Gene expression heatmap of the expressed *ClaSWEET* genes in the root and stem tissues of PI (PI296341-FR) and Sg (Sugar baby) after being infected with Fusarium wilt. Sg Fon R (infected root samples of Sugar baby); PI Fon R (infected root samples of PI296341-FR); Sg Fon S (infected stems of Sugar baby); PI Fon S (infected stems of PI296341-FR). Red: upregulated. Green: downregulated. Gene clusters were generated by average linkage clustering method. (**b**) Relative expressions of *ClaSWEET7*, *15*, and *19-21* in the stem of Sg and PI. (**c**) Relative expressions of *ClaSWEET9*, *11*, *12*, and *16-22* in root tissue of Sg and PI.

**Figure 8 ijms-22-08407-f008:**
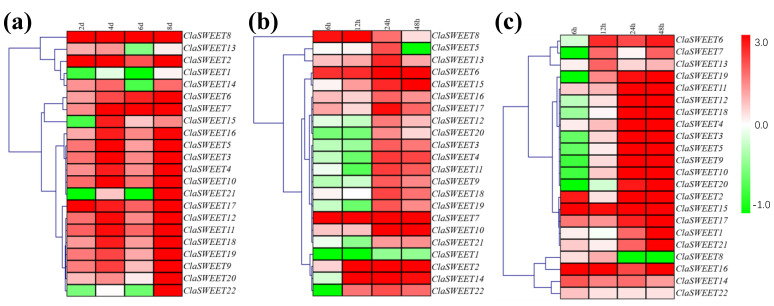
Gene expression heatmap of the *ClaSWEET* genes in the YL leaf under drought (**a**), salt (**b**), and low-temperature stress (**c**). The plant samples at 0 dpt and 0 hpt were considered as controls. Red: upregulated. Green: downregulated. The gene clusters were obtained with the average linkage clustering method.

**Table 1 ijms-22-08407-t001:** Characteristics of the *ClaSWEET* genes in watermelon.

*Arabidopsis* Homologous	Gene Name	Gene ID	Location (Strand)	CDS (bp) ^1^	TMDs ^2^	Protein Length (aa)	*p*I ^3^	MWs (kDa) ^3^	E-Value	Subcellular Localization ^4^
*AtSWEET3/AT5G53190.1*	*ClaSWEET1*	*Cla97C01G000640.1*	Chr:01 436923-440033 (−)	675	6	224	9.12	25.23	6.17 × 10^−83^	PM
*AtSWEET7/AT4G10850.1*	*ClaSWEET2*	*Cla97C01G001850.1*	Chr:01 1665964-1668736 (−)	795	6	264	9.83	29.1	3.96 × 10^−92^	PM
*AtSWEET5/AT5G62850.1*	*ClaSWEET3*	*Cla97C01G012030.1*	Chr:01 24835894-24837119 (+)	714	7	237	9.11	26.89	1.15 × 10^−105^	PM
*AtSWEET5/AT5G62850.1*	*ClaSWEET4*	*Cla97C01G012040.1*	Chr:01 24846574-24847102 (+)	207	1	68	4.85	7.66	5.33 × 10^−18^	EX
*AtSWEET5/AT5G62850.1*	*ClaSWEET5*	*Cla97C01G012050.1*	Chr:01 24854037-24854591 (+)	225	2	74	4.46	8.29	9.12 × 10^−26^	CY
*AtSWEET12/AT5G23660.1*	*ClaSWEET6*	*Cla97C03G055120.1*	Chr:03 4110250-4112328 (−)	885	7	294	8.36	33.02	6.10 × 10^−105^	PM
*AtSWEET12/AT5G23660.1*	*ClaSWEET7*	*Cla97C03G055130.1*	Chr:03 4123823-4125227 (−)	900	7	299	6.5	33.68	1.28 × 10^−108^	ER
*AtSWEET17/AT4G15920.1*	*ClaSWEET8*	*Cla97C03G067610.1*	Chr:03 31033031-31041340 (−)	879	7	292	9.3	32.13	2.66 × 10^−62^	PM
*AtSWEET17/AT4G15920.1*	*ClaSWEET9*	*Cla97C05G087740.1*	Chr:09 5868309-5872614 (−)	714	7	237	8.96	26.13	1.13 × 10^−73^	TM
*AtSWEET15/AT5G13170.1*	*ClaSWEET10*	*Cla97C06G111200.1*	Chr:06 1891143-1893938 (+)	804	7	267	9.02	30.26	8.57 × 10^−96^	PM
*AtSWEET12/AT5G23660.1*	*ClaSWEET11*	*Cla97C06G127900.1*	Chr:06 29274606-29275898 (−)	870	7	289	9.14	32.51	3.20 × 10^−95^	PM
*AtSWEET11/AT3G48740.1*	*ClaSWEET12*	*Cla97C06G127910.1*	Chr:06 29284604-29285549 (−)	645	5	214	8.65	23.91	2.97 × 10^−62^	PM
*AtSWEET7/AT4G10850.1*	*ClaSWEET13*	*Cla97C07G138130.1*	Chr:07 25774648-25778233 (+)	774	7	257	9.13	28.45	6.01 × 10^−95^	PM
*AtSWEET1/AT1G21460.1*	*ClaSWEET14*	*Cla97C08G149190.1*	Chr:08 17567657-17571032 (−)	756	7	251	9.25	27.629	3.61 × 10^−123^	PM
*AtSWEET4/AT3G28007.1*	*ClaSWEET15*	*Cla97C08G149470.1*	Chr:08 17854068-17855319 (−)	729	7	242	9.27	26.94	7.34 × 10^−95^	PM
*AtSWEET4/AT3G28007.1*	*ClaSWEET16*	*Cla97C10G197080.1*	Chr:10 26877106-26878414 (−)	729	7	242	8.95	27.05	1.07 × 10^−94^	PM
*AtSWEET4/AT3G28007.1*	*ClaSWEET17*	*Cla97C10G197130.1*	Chr:10 26899001-26900306 (+)	699	7	232	8.28	25.77	2.19 × 10^−88^	PM
*AtSWEET5/AT5G62850.1*	*ClaSWEET18*	*Cla97C10G197140.1*	Chr:10 26903294-26906670 (−)	579	5	192	8.53	21.99	9.81 × 10^−60^	PM
*AtSWEET5/AT5G62850.1*	*ClaSWEET19*	*Cla97C10G197150.1*	Chr:10 26918854-26921273 (−)	711	7	236	9.07	26.24	1.88 × 10^−95^	PM
*AtSWEET15/AT5G13170.1*	*ClaSWEET20*	*Cla97C11G216360.1*	Chr:11 17582166-17583939 (+)	342	2	113	9.73	13.06	2.60 × 10^−26^	CTM
*AtSWEET15/AT5G13170.1*	*ClaSWEET21*	*Cla97C11G216370.1*	Chr:11 17709586-17711235 (−)	324	2	107	5.82	12.63	1.91 × 10^−28^	CTM
*AtSWEET2/AT3G14770.1*	*ClaSWEET22*	*Cla97C11G223200.1*	Chr:11 29080934-29082856 (−)	702	7	233	9.22	26.02	1.07 × 10^−115^	PM

^1^ The length of coding domain sequences (CDS). ^2^ The number of transmembrane domains (TMDs). ^3^ The molecular weight (MWs) and theoretical isoelectric point (*p*I) were predicted using ‘Compute *p*I/Mw’ tool from ExPASy. ^4^ The subcellular localizations were predicted by WoLFPSORT. PM, plasma membrane; EX, extracellular; CY, cytoplasmic; ER, endoplasmic reticulum; TM, tonoplast membrane; CTM, chloroplast thylakoid membrane.

## Data Availability

The original contributions presented in the study are included in the article/Appendix A; further inquiries can be directed to the corresponding authors.

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
