# Peer review of "Systematic Genome-Wide Study and Expression Analysis of SWEET Gene Family: Sugar Transporter Family Contributes to Biotic and Abiotic Stimuli in Watermelon"

_ijms, 2021, doi:10.3390/ijms22168407_

Round 1

Reviewer 1 Report

Research questions are well defined and within the aims and the scope of the journal. Material and methods are suitable and properly described. Conclusions are well based on the results.

Text should be corrected by a native English speaker with scientific botanical and biochemical knowledge.

Other suggestions:

Line 15. Insert scientific name of watermelon here.

Line 19, instead of “gene”, better: “genes”.

Line 28, Insert scientific name of watermelon here as a keyword.

Line 50, instead of “A.thaliana”, here should be a correct genus name of the plant, and inserted a void place between genus and species name.

Lines 50-54, line 88 etc. scientific names of plants should be in italic.

Line 262, instead od “low-temperatur)”, correct “low-temperature)

Lines 569 – 576 should be completed with the information needed (for example ” NAME OF INSTITUTE (protocol code XXX and date of approval).”

Author Response

Research questions are well defined and within the aims and the scope of the journal. Material and
methods are suitable and properly described. Conclusions are well based on the results.

Answer: Thank you for supporting our research idea, materials and methods, results and discussion.

Text should be corrected by a native English speaker with scientific botanical and biochemical
knowledge.

Answer: Thank you for your kind suggestion. We agree that our manuscript was not appropriate from
language point. Hence, the current version is edited by professional person with genomic and
biochemical knowledge. The modifications included in track changes of revised manuscript.

Other suggestions:

Line 15. Insert scientific name of watermelon here.

Answer: Thank you for correction. We have added the scientific name, which appears to be important
correction. Line- 17

Line 19, instead of “gene”, better: “genes”.

Answer: As per suggestion term is modified. Line-24

Line 28, Insert scientific name of watermelon here as a keyword.
Answer: We appreciate the valuable suggestion. We agree that it is important suggestion and we have
added the scientific name of watermelon in keywords. Line-30

Line 50, instead of “A.thaliana”, here should be a correct genus name of the plant, and inserted a void
place between genus and species name.

Answer: Thank you for suggestion. We have made correction in revised version of manuscript. Line
53

Lines 50-54, line 88 etc. scientific names of plants should be in italic.

Answer: As per suggestion changes were made in revised version. Line -53-56, 990-92

Line 262, instead od “low-temperatur)”, correct “low-temperature)

Answer: Thank you for correction. We have revised the text as per suggestion. Line-265

Lines 569 576 should be completed with the information needed (for example” NAME OF
INSTITUTE (protocol code XXX and date of approval).”

Answer: Thank you for valuable suggestion. This part is not applicable in our manuscript so we have
made corrections as per requirements of journal. Methods and materials used in the study is are cited
with appropriate references.

Reviewer 2 Report

1. The authors need to improve the English language presented here and double check for spelling, orthography and grammar errors. I found several problems. So, I suggest going through someone with better English skills to fix these issues.

2. Results are mostly okay, however some figures need alteration and better explaining (in text and captions).

3. Figures and tables are not appropriate position in the text. 

4. Discussion is mostly appropriate.

5. Conclusion is good and properly reflects the results found here.

6. The authors should Italic all the gene names in the manuscript for example "The ClaSWEET genes were identified and named from ClaSWEET1 to ClaSWEET22 based on their appearance on the chromosomes, and each AtSWEET gene corresponds to approximately one to five ClaSWEET genes (Table 1).

Round 2

Reviewer 2 Report

I noticed a mistake with formatting in the MS PDF file; they did not upload the clean version of MS, therefore they will take care of MS when submitting the final version.